# Understanding the Burden of Legal Financial Obligations on Indigent Washingtonians

**Maria Katarina E. Rafael \* and Chris Mai** 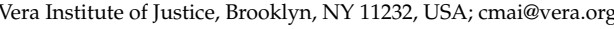

Vera Institute of Justice, Brooklyn, NY 11232, USA; cmai@vera.org
\* Correspondence: mrafael@vera.org

**Abstract:** In criminal courts across the country, judges assess a variety of fines, fees and other legal financial obligations (LFOs) that many defendants struggle to pay. This paper examines the disproportionate burden that fine and fee assessment and collection practices impose on low-income, system-involved individuals, using administrative court data for criminal cases filed in Washington's courts of limited jurisdiction between 2015 and 2020. The authors find that the majority of defendants do not or only partially pay their LFOs, but that these observations are more pronounced for indigent defendants. The authors also find that, of defendants who fully pay off their fines and fees, individuals with a public defender satisfy their debt after a greater number of days, as compared to individuals with private counsel. This is all in spite of public defender defendants generally being assessed smaller amounts in fines and fees at the outset. Additionally, the authors uncover that when defendants do pay off all of their fines and fees, they tend to do so on the day of assessment, with the likelihood of satisfying full payment generally decreasing as time goes on. These findings suggest that many people struggle with criminal justice debt, but that this problem disproportionately impacts indigent Washingtonians, subjecting them to a greater possibility of harm through the various methods of collections enforcement.

**Keywords:** fines; fees; legal financial obligations; debt; courts; criminal justice; indigence





## 1. Introduction

Criminal justice debt is a major problem in criminal courts across the country. Each day, these courts assess thousands of dollars in fines, fees, and other types of legal financial obligations (LFOs) that many defendants struggle to pay and risk never paying, at a devastating cost. Failure to pay LFOs can result in punishment by way of additional fees, the loss of driving privileges, arrest, and even jail time. Long-term, unresolved LFOs can ensnare debtors in an endless cycle of criminal legal system involvement not to mention, the impact of this debt on people's ability to maintain or attain housing, employment, and access to credit, all of which decrease the likelihood that they will ever be able to settle their criminal justice debt.

Given the severe consequences for non-payment of LFOs, investigation into fine and fee assessment and payment practices is merited to understand the scale of the debt burden on system-involved individuals. In this study, the authors specifically seek to understand who pays their LFOs and who does not, thereby taking on the often-insurmountable burden of criminal justice debt. Using attorney representation by a public defender as a proxy for indigence, the authors hypothesize that low-income individuals would be less able to afford court-imposed financial obligations, making them vulnerable to incur legal debt that they may never be able to address. To test this hypothesis, the authors use administrative data from the courts of limited jurisdiction in Washington State and evaluate whether an individual's indigence has a significant impact on their LFO assessment and payment outcomes. Because of the nature of the data, payment information is limited to just those

transactions within 180 days from assessment. See "Materials and Methods" section for more information.

In this report, the authors use administrative court data to examine the following questions:

- How much in LFOs is imposed on and paid (within 180 days) by defendants with a public defender as compared to defendants with a private attorney, on average?
- How are LFOs resolved, if ever, within 180 days of assessment for individuals with a public defender as compared to individuals with a private attorney?
- Of individuals who fully pay off their LFOs within 180 days, how long does it take individuals with a public defender to settle their debt as compared to individuals with a private attorney?
- What other case factors, in addition to attorney representation, influence LFO assessment and payment outcomes?

## 2. Literature Review

A number of studies on court systems around the country have demonstrated that many people do not pay the fines and fees that they owe, in spite of the numerous, serious consequences for not doing so, suggesting that many people struggle to afford these costs. Much of the research about inability to pay LFOs, however, has tended to take a holistic view of defendants rather than compare them based on indigency status (Harris and Edwards 2017; Martin et al. 2018).

Actually, the bulk of existing literature on criminal justice debt focuses on the impacts of this debt on successful reentry following a period of incarceration or community supervision. Findings from this body of research demonstrate that the majority of people reentering society owes substantial court debt and that this debt increases one's likelihood of remaining under supervision and ultimately returning to prison (Link 2017, 2019; Pleggenkuhle 2012). In the reentry context, researchers have determined that debt amounts are strongly associated with certain factors including income (Link 2019; Meredith and Morse 2007).

Fine and fee assessment and payment practices in Washington State specifically have been explored by a number of researchers over the past decade. Much of the published research about Washington centers on LFO assessments and collections at the felony level or within the state's superior courts (Beckett et al. 2008; Harris and Beckett 2010). The existing research on superior court debt reveals that the average debt possessed by people with felony convictions is substantial; and between 2004 and 2007 the majority of people with this debt paid $0 towards their assessed fines and fees, and only 12 percent paid between 91 and 100 percent of their assessed LFOs (Ibid.). One study, which does analyze LFO data for Washington's municipal courts, with a special focus on Seattle Municipal Court (SMC), found that for accounts filed in 2017 the majority of LFO assessments, even after significant downward adjustments by SMC, were still outstanding by the end of the year (Edwards and Harris 2020). Another study in Washington, which explores the issue of criminal justice debt in Pierce County's municipal, district, and superior courts, determined that for cases adjudicated in Pierce County between 2010 and 2017 only 36 percent of court debt was ever paid (Martin and Fowle 2019).

Another large portion of what is known about court debt comes from a handful of qualitative and some quantitative studies that have not been peer reviewed. In a survey of 20 families in Dane County, Wisconsin who had faced fines and fees from a child in the juvenile justice system, twelve were not paying at all and only two had fully paid what they owed (Paik and Packard 2019). A survey of nearly 1000 Alabamians facing fines and fees found that almost half believed that they would never be able to pay off what they owed, with many people forgoing basic necessities in order to make payments (Nelson et al. 2018). Studies of court data have found similar results. A study of individuals with cases in New Orleans municipal court who had pled guilty in the year 2015 found that by June 2016, only 29 percent had fully paid what they owed, 9 percent had made partial payments, and 62 percent had made no payments at all (Laisne et al. 2017). A study of Rhode Island's

felony Superior Court found that after four years, only half of people had fully paid off their debt with one-third still on a payment plan (Rhode Island Family Life Center 2008). The ACLU of Pennsylvania conducted one of the first empirical studies of court debt focusing on differences between indigent and non-indigent defendants, concluding that individuals represented by a public defender pay off a smaller share of their fines, costs, and restitution debt as compared to those represented by a private attorney, despite the fact that public defender defendants are assessed smaller amounts in LFOs to begin with (Ward et al. 2020). Oklahoma Policy Institute similarly found that 70 percent of all assessed LFOs, corresponding with assessments largely imposed on low-income Oklahomans, had never been paid (Shade 2020). They also determined that at a certain point criminal court collections plateau, inferring that there is a natural limit to how much the poorest Oklahomans are able to pay (Ibid.).

To date, no peer-reviewed work has empirically assessed differences in LFO assessments and payments between indigent and non-indigent defendants in Washington's courts of limited jurisdiction, which handle lower-level offenses and therefore frequently levy monetary sanctions on people with convictions. The authors believe that this paper will be a significant contribution to the field, advancing scholarship on LFOs broadly, but especially devoting critical attention to the impact of court debt on indigent defendants, who make up the majority of all people with criminal cases.[1]

## 3. Materials and Methods

The authors use case-level records for all criminal convictions in courts of limited jurisdiction in Washington State between July 2015 and November 2020 wherein a fine, fee, cost, or other legal financial obligation was imposed. Washington's courts of limited jurisdiction include district courts, which primarily handle misdemeanors and traffic violations; and municipal courts, which have authority over traffic and misdemeanor cases that originate within city limits, as well as local ordinance violations. The courts of limited jurisdiction tend to have higher fine and fee activity than superior courts.

These records, furnished by Washington's Administrative Office of the Courts (AOC), include data for all 61 district courts and 109 municipal courts across the state, except Seattle Municipal Court, which does not report data to AOC, and King County District Court, which does not report information for LFO payment dates to AOC. The data contain information corresponding with 614,803 unique dockets.

The variables available or constructed for the authors' case-level analysis include

- Court jurisdictional level (district*, municipal)[2],
- Defendant race (white*, non-white, refused or unknown)[3],
- Attorney representation (private attorney*, public defender)[4],
- Assessment date,
- Total assessment amount,
- Payment date(s)[5], and
- Total payment amount.

* Indicates reference group in multiple regression models

While authors conduct analysis at the case level, defendants can have multiple cases between 2015 and 2020. According to AOC data, approximately 21 percent of defendants have more than one conviction with financial sanctions, meaning they owe more than one set of LFOs. Additionally, the authors' analysis only captures information for cases with a single LFO assessment date. Cases with more than one assessment date tended to be more complicated in their formatting. These cases also typically signal the imposition of fees associated with surveillance and monitoring, whereas cases with a single assessment date typically only include conviction-related fines and fees.

The variable for total assessment amount factors in all upward and downward adjustments, making the number a net assessment. Downward adjustments are the result of waivers or credits applied by the court. Where downward adjustments resulted in full

compensation of LFOs, the authors treated the assessment as a $0 assessment.[6] Due to how they are reported in raw AOC records, assessments do not include restitution amounts. Payment data, however, do include restitution payments, although they are not recorded explicitly as such. Since there is no way to track or compare restitution payments to assessments, the authors treat restitution payment amounts as payments towards the fines and fees assessed.[7]

Additionally, due to the format in which the raw AOC records are recorded, it is difficult, if not entirely impossible, to accurately parse through payment data to understand exactly how long it takes an individual to pay off their criminal justice debt. That is, LFO payments are only recorded if they are paid within the same fiscal year as when the LFO was assessed. The authors apply a workaround by filtering the dataset to just those assessments within the first six months of the fiscal year and track payments made within six months (or 180 days) of the date of assessment.[8] This arrangement poses not an insignificant limitation to the ways in which the data can be interpreted, as it halves the number of observations in the authors' dataset and prohibits an analysis of payment or understanding of outstanding debt in the long term. Based on some of the research and findings outlined in the previous section, however, the authors have confidence to reasonably assume that much of the court debt left unresolved after 180 days will be left to linger for many more years. The authors also use this limitation of the data as an opportunity to test a secondary hypothesis: that people who have the ability to pay their LFOs will do so immediately, or on the day of assessment.

The authors retain only information for cases where data for all variables listed above are present in the records produced. Cases with assessments made in the last six months of a fiscal year (326,528) are excluded. The authors also exclude cases with multiple LFO assessment dates (38,795) or where any LFO payment date preceded the case's LFO assessment date (52,656). Cases with missing race information (1) and missing attorney representation data (85,653) were also excluded. The authors also exclude cases where charge information is missing (18,212).[9] All told, the authors' final dataset corresponds with information for 92,958 unique dockets. The authors acknowledge that trimming the dataset to this extent may lead to loss of information, decreased statistical power, and weakened generalizability of findings. See "Limitations" section for more information.

In this paper, the authors begin by summarizing mean and median assessments and payments (within 180 days of assessment). Then, the authors examine the distribution of LFOs that are fully paid, unpaid, partially paid, or otherwise settled (such as the relatively uncommon cases in which debt is eliminated through downward adjustments by the court) within 180 days of assessment. Next, the authors estimate the number of days it typically takes for defendants to fully settle their fines and fees, of defendants who manage to resolve their debt within 180 days. At each step, the authors compare outcomes for defendants represented by a public defender with those represented by private counsel.

Finally, the authors use general linear and logistic regression models in order to predict LFO assessment amounts, payment rates, and whether payment will be fully satisfied within 180 days of assessment. The authors include attorney representation, race, court jurisdictional level, and assessment amount as predictor variables, and plot regression models and odds ratios to interpret the models. All analyses are conducted in R version 4.0.3.

See Table 1 below for descriptive statistics summarizing all variables (except for datetime variables) included in the sample dataset.

**Table 1.** Descriptive statistics for non-datetime variables within sample dataset.

| Variable | N | Percentage of Cases | Mean | Std. Dev. | Min | Pctl. 25 | Pctl. 75 | Max |
|---|---|---|---|---|---|---|---|---|
| Court jurisdictional level | | | | | | | | |
|     District (reference) | 60,495 | 65% | - | - | - | - | - | - |
|     Municipal | 32,463 | 35% | - | - | - | - | - | - |
| Defendant race | | | | | | | | |
|     White (reference) | 64,628 | 70% | - | - | - | - | - | - |
|     Non-white | 25,392 | 27% | - | - | - | - | - | - |
|     Refused or unknown | 2938 | 3% | - | - | - | - | - | - |
| Attorney representation | | | | | | | | |
|     Private attorney (reference) | 17,073 | 18% | - | - | - | - | - | - |
|     Public defender | 75,885 | 82% | - | - | - | - | - | - |
| Total assessment amount | - | - | $620 | $749 | $0 | $150 | $843 | $75,000 ** |
| Total payment amount | - | - | $191 | $1133 | $0 | $0 | $150 | $200,000 ** |
| All observations | 92,958 | - | - | - | - | - | - | - |

** Payment amounts may exceed assessment amounts due to restitution. Restitution amounts are reflected in payments data but not in assessment data.

## 4. Results

### 4.1. Comparing LFO Assessment and Payment Practices for Individuals with a Public Defender as Compared to Individuals with a Private Attorney

This section details descriptive analyses of assessment and payment patterns across defendants with different attorney representation.

4.1.1. How Much in LFOs Is Imposed on and Paid (within 180 Days) by Defendants with a Public Defender as Compared to Defendants with Private Counsel, on Average?

As demonstrated in Table 2, the typical defendant with a private attorney is assessed several hundreds more in LFOs than the typical defendant with a public defender, and the typical defendant with a public defender pays less both in overall amount and proportionally out of the assessed amount within 180 days than the typical defendant with a private attorney. Because financial variables are not normally distributed, the authors applied a non-parametric test in order to assess differences in means between the two groups of defendants. The authors conducted Wilcoxon rank sum tests (also called Mann–Whitney U tests), finding that mean LFO assessments, payments, and rates of payments are statistically significantly different for private attorney defendants as compared to public defender defendants (*p*-value < 0.001 for each test).

**Table 2.** Mean and median LFO assessments, payments within 180 days of assessment, and rates of payment within 180 days of assessment.

| | Private Attorney Defendants | Public Defender Defendants | All Defendants |
|---|---|---|---|
| N (number of cases) | 17,073 | 75,885 | 92,958 |
| Mean assessed | $773 | $585 | $620 |
| Mean paid within 180 days | $376 | $149 | $191 |
| Mean percent paid within 180 days[10] | 43% | 19% | 23% |
| Median assessed | $550 | $350 | $393 |
| Median paid within 180 days | $150 | $0 | $0 |
| Median percent paid within 180 days[11] | 21% | 0% | 0% |

The higher assessment amounts for private attorney defendants suggest that judges may be taking indigency status into consideration when assessing fines and fees at conviction. The lower rates of LFO payment for defendants with a public defender, in spite of already lower assessment amounts, suggest that these defendants have more difficulty paying their LFOs. This finding comports with the authors' hypothesis that individuals with a public defender have less access to wealth than individuals who hire a private attorney for legal representation, influencing the amount they are able to contribute towards their LFOs. It should be noted, however, that LFO payment rates are quite low for both groups.

### 4.1.2. How Are LFOs Resolved, If Ever, within 180 Days of Assessment for Individuals with a Public Defender as Compared to Individuals Represented by a Private Attorney?

Only a very modest share (18 percent) of all defendants fully pays off its LFOs within 180 days of assessment, but the share is even smaller for individuals with a public defender (15 percent, Table 3), especially when compared to their counterparts represented by a private attorney (35 percent, Table 3).[12] In order to test the null hypothesis that the proportions (or probabilities of success) in the two groups of defendants are the same, the authors conducted a two-sample test for equality of proportions. The authors find that the proportion of cases for which LFOs were fully paid within 180 days of assessment is statistically significantly lower for individuals with a public defender as compared to those with a private attorney ($p$-value < 0.001).

**Table 3.** Distribution of cases by LFO resolution or status within 180 days of assessment[13].

|  | Private Attorney Defendants | Public Defender Defendants | All Defendants |
| --- | --- | --- | --- |
|  | N (Pct) | N (Pct) | N (Pct) |
| Fully paid LFOs on day of assessment | 1323 (8%) | 1908 (3%) | 3231 (3%) |
| Fully paid LFOs between 1 and 180 days after assessment | 4545 (27%) | 8940 (12%) | 13,485 (15%) |
| Partially paid LFOs within 180 days of assessment | 4508 (26%) | 13,171 (17%) | 17,679 (19%) |
| Made zero payments towards LFOs within 180 days of assessment | 6493 (38%) | 51,575 (68%) | 58,068 (62%) |
| LFOs fully compensated through credits or waivers | 204 (1%) | 291 (0.38%) | 495 (1%) |
| TOTAL | 17,073 (100%) | 75,885 (100%) | 92,958 (100%) |

An approximately equivalent share (19 percent, Table 3) of all defendants partially pay their LFOs within 180 days of assessment. This share is much smaller for public defender defendants (17 percent, Table 3) than private attorney defendants (26 percent, Table 3). A two-sample test for equality of proportions finds that the proportion of cases for which LFOs were partially paid within 180 days of assessment is statistically significantly lower for individuals with a public defender as compared to those with a private attorney ($p$-value of <0.001).

Strikingly, the majority of defendants (62 percent) do not make a single payment towards their LFOs within 180 days of assessment. By this metric, individuals with a public defender are less likely to make any payment towards their LFOs within those 180 days. A two-sample test for equality of proportions finds that the proportion of cases for which zero LFO payments were made within 180 days of assessment is higher for individuals with a public defender as than for those with a private attorney (results are statistically significant with a $p$-value of <0.001).

### 4.1.3. Of Individuals Who Fully Pay off Their LFOs within 180 Days, How Long Does It Take Individuals with a Public Defender to Settle Their Debt as Compared to Individuals with a Private Attorney?

It takes approximately 55 days (Table 4), on average, or a median of 35 days (Table 4), for individuals to fully pay off their LFOs. Individuals with a public defender tend to fully satisfy their court debt after a greater number of days than individuals with a private attorney. The authors conducted a Wilcoxon rank sum test, finding that the mean number

of days until LFOs are fully paid is statistically significantly greater for public defender defendants than for private attorney defendants (*p*-value of < 0.001). The most frequent number of days it takes until LFOs are fully paid off for both groups of defendants is zero, meaning that full payment is made on the day of assessment.

**Table 4.** Mean, median, and mode number of days until LFO debt is settled (of cases where LFO debt is settled within 180 days).

|  | Private Attorney Defendants | Public Defender Defendants | All Defendants |
|---|---|---|---|
| N (number of cases that have fully paid LFOs within 180 days) | 5,868 | 10,848 | 16,716 |
| Mean number of days until LFOs are paid off | 44 | 61 | 55 |
| Median number of days until LFOs are paid off | 25 | 45 | 35 |
| Mode number of days until LFOs are paid off | 0 | 0 | 0 |

Table 5 shows the scale of cases involving individuals wherein the number of days until LFOs are fully paid off is zero or wherein all LFOs are paid in full on the day of assessment, of only those cases where LFOs are fully paid within 180 days. In total, 19 percent of individuals who fully pay their LFOs pays them off on the day their LFOs are assessed. Even of those who fully pay of their LFOs within 180 days, public defender defendants are less likely than their private attorney counterparts to pay their LFOs immediately. A two-sample test for equality of proportions finds that the proportion of cases for which LFOs were paid in full on the day of assessment (of cases where LFOs were fully paid within 180 days of assessment) is greater for individuals with a private attorney as compared to those with a public defender (results are statistically significant with a *p*-value of <0.001).

**Table 5.** Numbers and percentages of cases involving defendants who fully satisfy LFO payment on the day of assessment.

|  | Private Attorney Defendants | Public Defender Defendants | All Defendants |
|---|---|---|---|
| N (number of cases that have fully paid LFOs within 180 days) | 5868 | 10,848 | 16,716 |
| Number of cases involving defendants who pay off LFOs on the day of assessment | 1323 | 1908 | 3231 |
| Percent of cases involving defendants who pay off LFOs on the day of assessment | 23% | 18% | 19% |

Figure 1 is a mirrored histogram demonstrating the range and distribution of the number of days until private attorney and public defender defendants (with observations for public defender defendants displayed in the negative) fully satisfy their court debt. Histogram bins are organized in the following increments: 0 (the day of assessment), 7, 14, 30, 60, 90, 120, 150, and 180 days. All observations corresponding with an increment of time greater than 180 days represent cases wherein full payment was not achieved within 180 days.

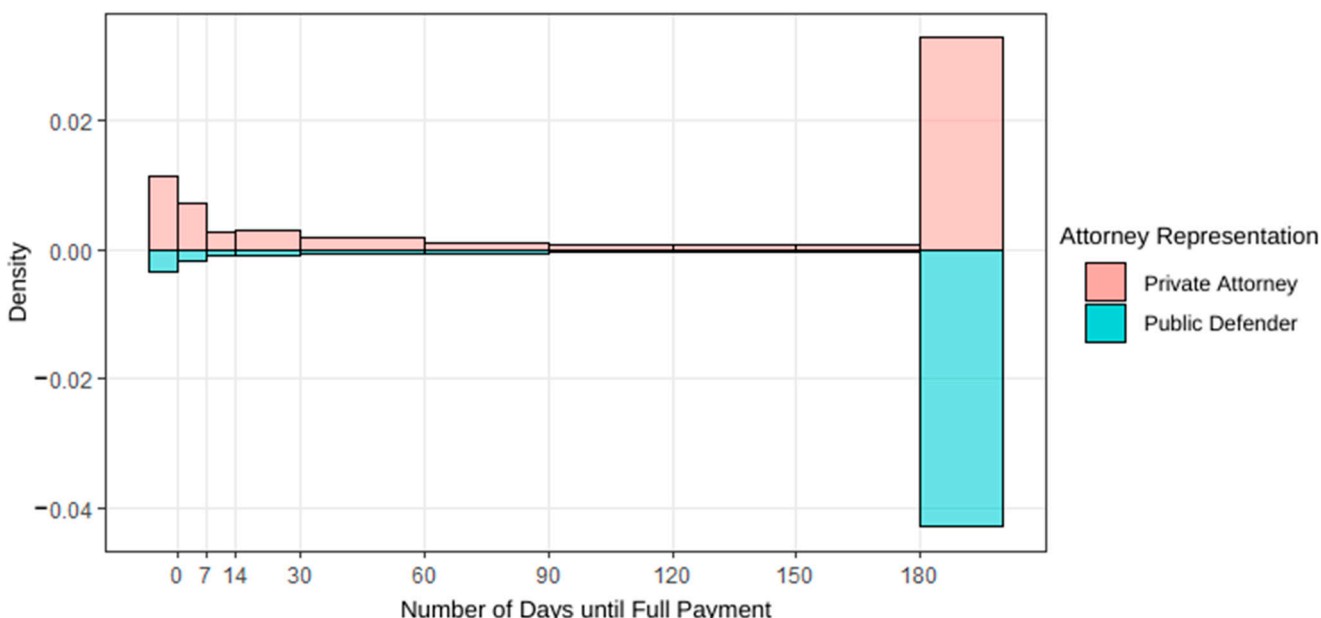

**Figure 1.** Distribution of the number of days until LFO debt is fully satisfied.

Consistent with the findings above, these back-to-back histograms reveal that proportionally more individuals with a private attorney fully pay off their LFOs within 180 days of assessment, when compared to individuals with a public defender. Additionally, private attorney defendants tend to satisfy payment earlier than their public defender counterparts. These plots also show that, for both groups of defendants but especially for defendants with a public defender, LFO payment decreases substantially several days out from assessment and essentially plateaus several months out from assessment. Combined, these findings suggest that the people who have the ability to pay their LFOs will pay and do so immediately, while the people who do not have the ability to pay suffer with the burden of criminal justice debt for at least half of a year and—as other research suggests—possibly for much longer than that.

### 4.2. Multiple Regression Analyses of LFO Assessment and Payment Practices

To explore how not only attorney representation but also other case factors impact LFO assessment and payment practices, the authors construct a series of regression models. The authors examine whether attorney representation, race, and court jurisdictional level influence assessment amounts (Model 1); the rate of LFO payment within 180 days of assessment (Model 2);[14] and whether full payment is made within 180 days of assessment (Model 3).[15]

Amount assessed = $\beta 0$ + $\beta 1$ Attorney Representation + $\beta 2$ Race + $\beta 3$ Court Jurisdictional Level + $\varepsilon$, (Model 1, Linear Regression)

Rate of LFO payment = $\beta 0$ + $\beta 1$ Attorney Representation + $\beta 2$ Race + $\beta 3$ Court Jurisdictional Level + $\beta 4$ Amount Assessed + $\varepsilon$, (Model 2, Linear Regression)

LFOs fully paid within 180 days = $\beta 0$ + $\beta 1$ Attorney Representation + $\beta 2$ Race + $\beta 3$ Court Jurisdictional Level + $\beta 4$ Amount Assessed + $\varepsilon$, (Model 3, Logistic Regression)

Figures 2 and 3 are graphical representations of the associations between outcomes (1) and (2) and the case factors outlined above, utilizing regression coefficients resulting from general linear regression models 1 and 2 outlined above.[16] The left-hand column shows the predictors of interest, and the regression coefficients and confidence intervals (95%) are plotted on the right. The location of the dots on the x-axis corresponds with the direction and size of the association between the outcome and the predictor. The line through the dot represents the size of the confidence interval, with a longer line representing greater uncertainty about whether the association is due to chance. Dots farthest left from the

x-intercept at 0 signify predictors most strongly associated with the outcome tested, in the negative, while dots farthest right signify variables most strongly associated with the outcome, in the positive.

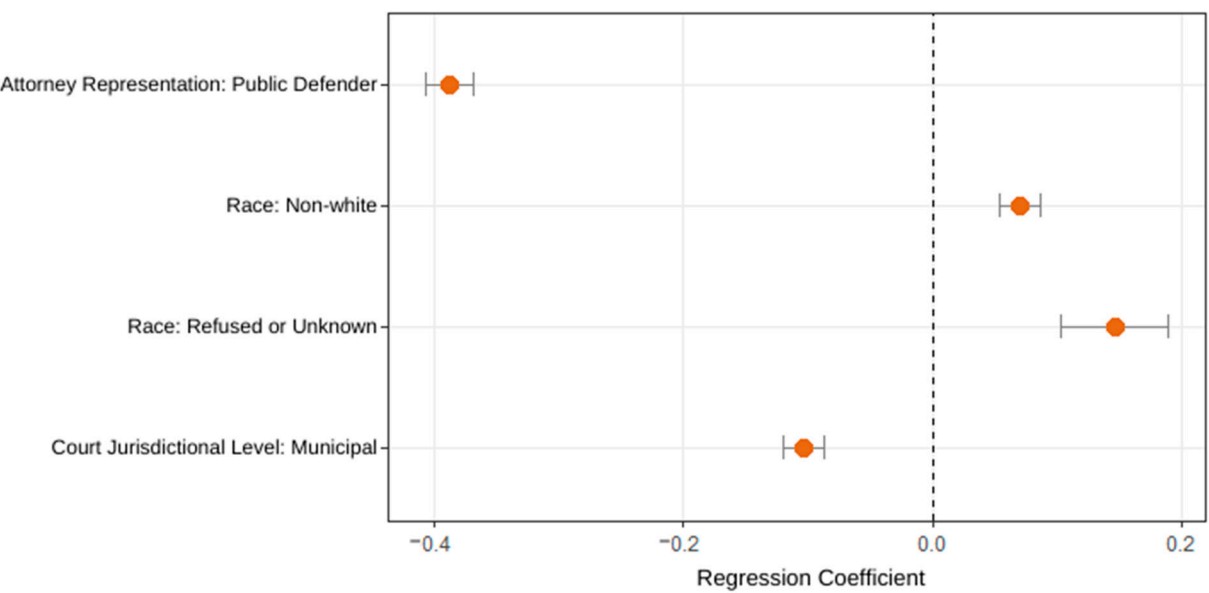

**Figure 2.** Regression coefficients for the various predictors of LFO assessment.

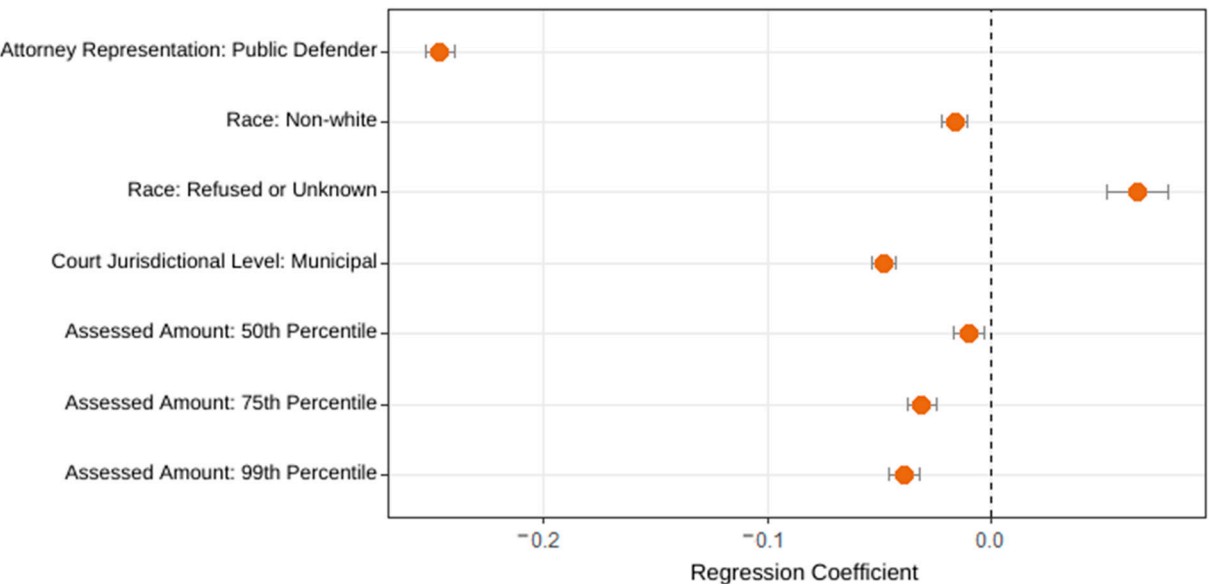

**Figure 3.** Regression coefficients for the various predictors of a higher rate of LFO payment.

Figure 4 is a graphical representation of the association between outcome (3) and the case factors outlined above, using odds ratios (ORs) resulting from the logistic regression model outlined above. As with the two previous figures, the left-hand column shows the predictors of interest. The ORs and confidence intervals (95%) are plotted on the right. The location of the dots on the x-axis corresponds with the direction and size of the association between the outcome and the predictor, and the line through the dot represents the size of the confidence interval. Dots farthest left from the x-intercept at 1 signify variables predicting lower odds that defendants will fully satisfy LFO payment within 180 days, while observations farthest right signify variables predicting higher odds that they will resolve their court debt.

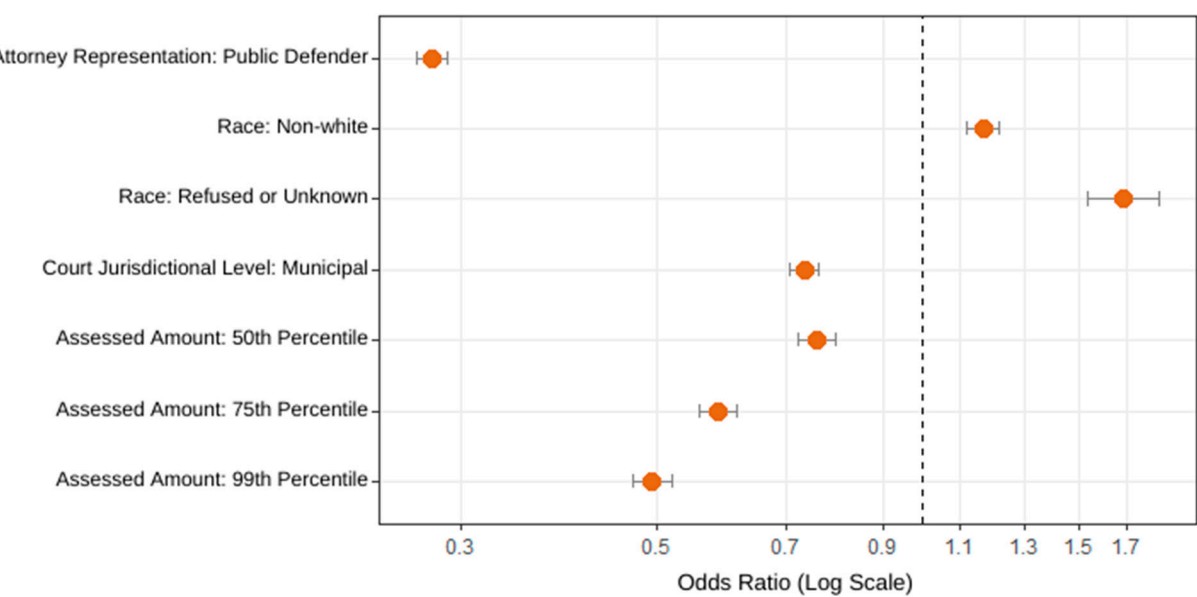

**Figure 4.** Odds ratios for the various predictors of full LFO payment within 180 days of assessment.

### 4.2.1. Attorney Representation

According to these models, whether an individual is represented by a public defender as compared to a private attorney is associated with LFO assessment and payment outcomes. The amount of LFOs assessed and whether a defendant is represented by a public defender have an inverse relationship, though the magnitude of the association is somewhat small (b = −0.388, *p*-value < 0.001, Figure 2). This finding suggests that indigent defendants are generally issued smaller amounts in fines and fees and that judges may be taking indigency status into consideration when assessing LFOs at conviction. In spite of public defender defendants being imposed lower assessment amounts, however, they tend to have lower LFO payment rates than their private attorney counterparts (the magnitude of this association is also small with a b of −0.246, *p*-value < 0.001, Figure 3), and the odds that they will fully pay off their court debt within 180 days is low (OR = 0.279, *p*-value < 0.001, Figure 4). These findings comport with the results from the authors' descriptive analyses outlined in the previous section.

### 4.2.2. Race

Race is also a predictor for LFO assessment and payment outcomes, although to a lesser degree than attorney representation. Assessment amounts and whether a defendant is non-white are only very minorly associated (b = 0.070, *p*-value < 0.001, Figure 2). Defendants of color have slightly lower payment rates than white defendants (the magnitude of this association is very small with a b of −0.017, *p*-value < 0.001, Figure 3) and are less likely to settle their LFO debt within 180 days (OR = 0.846, *p*-value < 0.001, Figure 4). Race as a case factor then appears to function similarly to attorney representation, with defendants of color also experiencing some of the same difficulties paying off their fines and fees as defendants with public defenders. Whether a defendant's race information is refused or unknown is also an important predictor for LFO assessment and payment outcomes. This finding raises questions about the missingness of race data in the dataset (2938 cases or 3 percent of all cases analyzed had race information marked as "refused or unknown"). The associations between race and LFO assessment and payment outcomes, however, are expressed to a lesser degree than the association between attorney representation and assessment and payment outcomes.

### 4.2.3. Court Jurisdictional Level

The relationship between LFO assessment and whether the court issuing the assessment is a municipal, rather than district, court is very weak and negatively associated (b = −0.103, *p*-value < 0.001, Figure 2). According to Model 2, defendants with cases in municipal court may have lower rates of LFO payment (the magnitude of this association is also very small with a b of −0.050, *p*-value < 0.001, Figure 3), and have lower odds that their LFOs will be fully satisfied within 180 days of assessment (OR = 0.711, *p*-value < 0.001, Figure 4) when compared to individuals who owe debt to district courts.

### 4.2.4. Amount Assessed

The authors include the amount assessed in LFOs as a predictor variable for models two and three to understand whether assessments have a noticeable impact on payment outcomes. In order to mitigate overdispersion, the authors created a new variable for assessment amount, categorizing the amounts by quartiles (the reference is 25th percentile assessment amounts). According to the models, assessment amounts have a modest impact on the rate of a defendant's LFO payment, with higher assessments being associated with lower rates of payment (for 50th percentile assessment amounts: b = −0.017, *p*-value = 0.004; for 75th percentile assessment amounts: b = −0.038, *p*-value < 0.001; for 99th percentile assessment amounts: b = −0.005, *p*-value < 0.001; Figure 3). Additionally, the odds of defendants settling their court debt within 180 days also do vary only modestly based on assessment amount for 50th percentile assessment amounts: OR = 0.758, *p*-value = 0.004; for 75th percentile assessment amounts: OR = 0.586, *p*-value < 0.001; for 99th percentile assessment amounts: OR = 0.494, *p*-value < 0.001; Figure 4). The association between payment outcomes is more strongly associated, however, with whether a person has a public defender rather than the amount in LFOs assessed, dispelling the notion that higher assessment amounts primarily drive worse payment outcomes.

## 5. Discussion

This study sought to assess the extent to which indigent Washingtonians experience a disproportionate burden from legal financial obligations. Consistent with existing literature, results demonstrated that defendants with lower incomes are issued lower amounts in LFOs, indicating that judges may be using their discretion to impose lesser sanctions on people they believe have less ability to pay (Link 2019; Ward et al. 2020). Additionally, in line with the other research on this topic, the results of this study suggest that despite being assessed lower amounts in LFOs to begin with, low-income defendants still experience great difficulty paying off their debt, and that even LFO amounts that may not appear overly burdensome (according to the authors' sample, an average amount of $585 and median of $350 for public defender defendants) are still near-impossible for indigent defendants to address (Ibid.).

Additionally, the administrative data the authors analyzed reveal that successful repayment of criminal justice debt becomes less and less likely as time goes on, in accordance with the few, recent publications that study LFO payments over time (Edwards and Harris 2020; Martin and Fowle 2019; Shade 2020; Ward et al. 2020). The authors found that while this pattern is true for all defendants, it is more pronounced for individuals who are indigent, and even for seemingly very low amounts of debt that are associated with lower-level offenses. Findings also suggest that when people do satisfy their LFOs, they typically pay immediately—or on the day of assessment. As such, this study contributes some new results to a small, but growing body of research.

The authors also found that, when compared to other case factors, even including the amount of LFOs assessed, whether a person had a public defender emerged as the strongest, significant variable predicting lower odds of full payment and higher odds of greater time needed to resolve outstanding debt. To the authors' knowledge, it has yet to be reported in the quantitative literature that indigence is so influential relative to other factors, including defendant race, court jurisdictional level, and assessment amount.

Taken together, these findings underscore that indigent defendants will continue to be destabilized by the burden of criminal justice debt unless steps are taken to reduce this burden. Many states, including Washington, have done much to reform sentencing of LFOs but still permit the assessment of certain mandatory conviction fees, leaving costs to be paid that people with low incomes consider substantial. Since study results showed that judges may already be using the full extent of their discretion to calibrate monetary sanctions to the financial means of defendants, potential policy solutions would require going further to eliminate any conviction fees that are statutorily mandated or expand judicial discretion to waive or reduce these fees based on a determination of indigence. Such reforms would do much to relieve harm for the overwhelming share of people who simply have no means to address their court debt, and by relying on a determination of indigence as a proxy for inability to pay would eliminate any additional burden of process created by a separate inquiry into ability to pay.

## 6. Limitations

The findings of this study are qualified by a set of limitations. As outlined in the "Materials and Methods" section, the data are not the product of random sampling strategy, but rather constructed through a set of narrow inclusion criteria due to missing and/or possibly erroneous data in the raw records. These inclusion criteria reduce the number of unique dockets captured in the dataset from 614,803 to just 92,958, or 15 percent of the original set of dockets with information provided by the AOC. Therefore, the sample is not perfectly representative of the entire universe of cases and may contribute to an incomplete or biased understanding of the data analyzed.

For example, by limiting analysis to just those cases with assessments issued within the first six months of a given fiscal year, the authors may be overlooking patterns associated with differences in case types, assessments, and payments during certain times of the year. Similarly, because data from the AOC do not include any Seattle Municipal Court data or King County District Court (KCDC) payment date data, the authors may be losing important information related to cases and convictions in the two courts situated in the state's most populous county. SMC and KCDC possess some of the highest caseloads among all of Washington's courts of limited jurisdiction.[17]

Additionally, the authors excluded information for cases with more than one LFO assessment date. Cases with a single LFO assessment date typically only include conviction-related fines and fees, whereas cases with multiple LFO assessment dates tend to include fees associated with one-time or recurring costs for engaging with services as a part of diversion and community supervision programming. As a result, the authors' sample may be underestimating total costs for both private attorney and public defender defendants and ignoring potential differences in these groups' assessments and payments related to pre-adjudication surveillance and monitoring fees.

Similarly, exclusion of cases with missing, refused, or unknown race, attorney representation, and charge information raises concerns about possible dilution of the sample data. After running a sample comparison to test for potential biasing effects, the authors found that results from descriptive analyses using the data with missing, refused, or unknown race information appeared to be reasonably similar to those that with complete race data. That is, the same dichotomy between public defender and private attorney defendants was prevalent in both the sample of data with missing, refused, or unknown race information as well as the sample of data with known race information. In the dataset with missing, refused, or unknown race information, however, LFO assessments were generally a bit higher and so were LFO payments within 180 days. It is unclear why assessment and payment amounts differed in this way.

Regarding information about attorney representation, missing data may be the result of record-keeping errors (i.e., the defendant did have attorney representation but it was not recorded for whatever reason), or the result of the person actually having no attorney representation. Unfortunately, the raw records provide no way of indicating which reason

resulted in the incomplete attorney representation field. A sample comparison testing differences between the sample of data with missing attorney representation data and the final sample data (with complete attorney representation data) reveals that LFO assessments for the former sample are much lower than for the latter sample. This suggests that many or most of the defendants with missing attorney representation info may have in fact been unrepresented, since lower fine amounts can be associated with lower-level offenses, such as traffic tickets, where attorney representation would not be necessary or typical. In fact, a review of the sample data with missing attorney representation data does confirm that the majority of the convictions within this sample were for lower-level, traffic-related charges.

While the authors excluded dockets with incomplete or missing charge information from our analysis, the authors did not include charge as a variable for analysis. This is because of possible overfitting due to the number of unique charges being relatively large in comparison to the sample size. The authors did, however, isolate out information for dockets corresponding with the most common charge (Driving While License Suspended in the Third Degree or "DWLS3," which made up 13 percent of all convictions represented in the sample dataset), to test whether the same findings would hold true across all dockets of a single charge type. Typically, a DWLS3 charge occurs when a driver receives a ticket for a moving violation and fails to comply with deadlines to pay the fines and fees associated with the ticket or appear in court to contest the ticket (Smith 2018). When examining just those data corresponding with LFOs assessed and paid for DWLS3 convictions, the same findings generally hold true. The authors conduct the same Wilcoxon rank sum tests and two-sample tests for equality of proportions, yielding results consistently in the identical direction as results corresponding with the tests conducted on the dataset including all cases. The results corresponding with just DWLS3 convictions, however, vary in their levels of significance.[18]

## 7. Conclusions

In summary, this study provides some new insights into the burden of LFO assessments on low-income defendants. This paper's novel contributions are its finding that indigence is associated with lower assessment amounts, lower odds of full repayment of assessed LFOs, and longer periods of carrying unpaid debt; as well as that the largest share of people who make full payment towards their LFOs does so on the day of assessment. Altogether, these results suggest that within Washington's courts of limited jurisdiction, indigent defendants struggle the most with criminal justice debt, indicating that the many harms associated with failure to pay fines and fees fall disproportionately on Washingtonians with the lowest incomes. The authors' policy recommendations highlight the need for greater ability to waive or reduce LFOs. Like other work of this type, this study raises concerns over fairness of statutorily mandated LFOs and defendants' ability to pay.

Future research by the authors will incorporate data that would permit analysis of long-term payments, beyond just six months. Additional research will also include data from Seattle Municipal Court and King County District Court and feature a cluster analysis to reveal more about who the typical public defender defendant is and what specific types of fines and fees most impact their ability to satisfy outstanding debt.

**Author Contributions:** Conceptualization, M.K.E.R. and C.M.; methodology, M.K.E.R.; validation, M.K.E.R.; formal analysis, M.K.E.R.; resources, M.K.E.R. and C.M.; data curation, M.K.E.R.; writing—original draft preparation, M.K.E.R.; writing—review and editing, M.K.E.R. and C.M.; visualization, M.K.E.R.; supervision, C.M.; project administration, M.K.E.R. and C.M.; funding acquisition, C.M. All authors have read and agreed to the published version of the manuscript.

**Funding:** This research was funded by Arnold Ventures and the Vera Institute of Justice. Views expressed in this article are those of the authors.

**Institutional Review Board Statement:** IRB Project No. IRB00001967 was approved by the Vera Institutional Review Board (Org. No. ORG0001520) on 9 February 2021.

**Informed Consent Statement:** The IRB granted a waiver of informed consent for this research based on their findings that: the research presented no more than minimal risk to participants, the waiver would not adversely affect the rights and welfare of the subjects, and the research could not practicably be carried out without the waiver or alteration. The data researchers requested contained information corresponding with thousands of individuals, without names, contact information or other information that would make it easy to obtain consent. If that information were to have been collected, the process, in itself, would be both impracticable and would create more issues with confidentiality.

**Data Availability Statement:** The administrative court data were produced by the Washington State Administrative Office of the Courts on 20 February 2021. Because these data contain personal identifiable information, per the requirements of the authors' memorandum of understanding regarding data sharing, the authors are restricted from making the raw data analyzed in this study available to persons other than the authors and their research team.

**Acknowledgments:** The authors would like to thank Frankie Wunschel, Abdelhamid Arbab, and Shefali Das for their contributions to this research and analysis.

**Conflicts of Interest:** The authors declare no conflict of interest.

## Notes

[1] According to the National Legal Aid and Defender Association (NLADA), approximately 80 percent of criminal defendants cannot afford a lawyer. See National Legal Aid and Defender Association. November 2020. "National Legal Aid & Defender Association Recommendations to the Biden-Harris Administration." Available online: https://www.nlada.org/sites/default/files/NLADA%20Recommendations%20to%20the%20Biden-Harris%20Administration.pdf (accessed on 3 November 2021).

[2] To avoid a sample size issue and possible overfitting, the authors use court jurisdictional level, rather than court, as a predictor variable.

[3] The authors categorize all defendants with "Hispanic" ethnicity as "non-white" race for purposes of this analysis. The Washington State Commission on Hispanic Affairs uses the terms "Hispanic" and "Latino" interchangeably. Washington State Commission on Hispanic Affairs. "WA State Demographics." Available online: https://www.cha.wa.gov/demographics-washington-state (accessed on 11 August 2021).

[4] Since AOC data does not consistently include whether a defendant was represented by a public or private attorney, the authors manually collected data on the employment history of each lawyer, in order to classify them as public defenders or private attorneys (data span cases from 2015 to 2020, so former public defenders were only labeled as such if they were practicing during that time period). To do this, the authors cross-refenced data with existing Washington Public Defender staff directories as well as other public databases (e.g., the membership list for the Washington Defender Association, an organization dedicated toward public defender trainings and reform; the Vancouver Defenders; and different legal aid organizations like the Northwest Justice Project), but this was by no means fully exhaustive; only certain counties maintain their directories, and many rely on contracted law offices for their public defense. As such, the authors manually searched for each lawyer's employment history on sites like Avvo and LinkedIn, amongst others, to establish their status as public versus private attorneys. One limitation was the lack of information on firms contracted for public defense. To ameliorate this, the authors looked at client reviews, county budgets, and news announcements, for signs of contracted work. Additionally, an unusually high caseload (greater than 1000 cases) gave indication that an attorney was either a public defender or a prosecutor, and the latter was easy to identify through an employment search. This resulted in the classification of most attorneys represented in the data as either public defenders or private attorneys, with some being classified as prosecutors, judges, or city attorneys, and few left unidentified (N/A). Some cases indicated multiple attorneys; where a defendant had at least one public defender represent them, the authors treated the case as a public defender case. The authors retained only information for cases where the authors could identify attorney representation.

[5] More than one LFO payment date signals payment in installments. In some instances, LFO payments appear to have been made on dates preceding the associated LFO assessment. These entries are treated as erroneous, and the authors exclude all cases for which any payment was made on a date preceding assessment.

[6] Cases with $0 assessments are generally excluded from analysis unless otherwise indicated.

[7] For example, if for a particular case fines and fees were assessed at $100 and restitution was assessed at $50 and the authors observed a singular payment of $125, the authors would only be able to discern that $100 was assessed and $125 was collected. The authors would treat such payment as payment in full, even though there is an outstanding $25 in restitution. If for a different case, fines and fees were assessed at $100 and restitution was assessed at $50, and the authors observed two payments of $50 each (assume the person paying meant for one payment to go towards restitution and the other towards their fines/fees), the authors would only be able to discern that $100 was assessed and $100 was collected. The authors would also treat these payments as payment in full, even though there is an outstanding $50 in fines and fees.

[8] The authors are pursuing additional data that would permit analysis of long-term payments for a future report.

9   The authors do not include charge as a variable for analysis because of possible overfitting due to the number of unique charges being relatively large in comparison to the sample size.

10  This refers to the average of all individual payment rates or percentages paid per case.

11  This refers to the median of all individual payment rates or percentages paid per case.

12  The authors conducted a two-sample test for equality of proportions, finding that the proportion of cases for which LFOs were fully paid within 180 days of assessment were different for individuals with a private attorney as compared to those with a public defender (results are statistically significant with a *p*-value of <0.001).

13  Percentages may not exactly total 100% due to rounding.

14  This regression model also includes assessment amount as a predictor variable.

15  Ibid.

16  For Model 1, which predicts assessment, the authors apply a log transformation to normalize the distribution in preparation for a standard linear regression.

17  According to the Washington Courts of Limited Jurisdiction Annual Caseload Report for 2020, Seattle Municipal Court had the highest caseload of any Washington municipal court, possessing 26 percent of all caseloads within Washington's courts of limited jurisdiction. King County District Court had the third-highest caseload of any Washington district court, possessing 3% of all caseloads in Washington's courts of limited jurisdiction. Courts of Limited Jurisdiction Annual Caseload Report for 2020. Washington Courts. Available online: https://www.courts.wa.gov/caseload/?fa=caseload.showReport&level=d&freq=a&tab=&fileID=rpt01 (accessed on 15 November 2021).

18  The authors conducted Wilcoxon rank sum tests to test group differences, finding that the average LFO assessment for DWLS3 convictions is greater for private attorney defendants than for public defender defendants (results are, however, not statistically significant, with a *p*-value of 0.14), and that the average rate of LFO payment for DWLS3 convictions is greater for private attorney defendants than public defender defendants (results are statistically significant with a *p*-value of < 0.001). The authors conducted 2-sample tests for equality of proportions, finding that the proportion of cases for which LFOs were fully paid within 180 days of assessment were greater for individuals with a private attorney as compared to those with a public defender (results, however, are not statistically significant, with a *p*-value of 0.49); the proportion of cases for which LFOs were partially paid within 180 days of assessment were different for individuals with a private attorney as compared to those with a public defender (results are statistically significant with a *p*-value of < 0.001); and the proportion of cases for which no LFO payments were made within 180 days of assessment were different for individuals with a private attorney as compared to those with a public defender (results are statistically significant with a *p*-value of < 0.001). The authors conducted a Wilcoxon rank sum test to test group differences, finding that mean number of days until LFOs are fully paid is greater for public defender defendants than for private attorney defendants (results are statistically significant with a *p*-value of < 0.001). The authors conducted a 2-sample test for equality of proportions, finding that the proportion of cases for which LFOs were paid in full on the day of assessment (of cases where LFOs were fully paid within 180 days of assessment) were greater for individuals with a private attorney as compared to those with a public defender (results, however, are not statistically significant, with a *p*-value of 0.65).

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
