# Peer review of "Understanding the Burden of Legal Financial Obligations on Indigent Washingtonians"

_socsci, doi:10.3390/socsci11010017_

Round 1

Reviewer 1 Report

Basic findings that are not surprising but helpful to know. Paper ignores all research previously done in WA (and of all states this is the bulk of research that exists on LFOs) - so surprising no reference. Also, need more clarity on removed data due to race missing.  Authors should engage with research using same AOC data and those that used Seattle Muni data.  How do findings compare?

Author Response

See PDF document attached.

Reviewer 2 Report

This is an important contribution to the field. My only suggestion would be to explain why the PA, AL, and OK studies - while useful - are not sufficient to prove the proposition studied here.

Author Response

See Word document attached.

Reviewer 3 Report

Thank you for the opportunity to review the manuscript “Understanding the Burden of Legal Financial Obligations on Indigent Washingtonians”. This paper leveraged unique administrative data from court records to examine repayment of assessed LFOs among indigent defendants in Washington’s courts of limited jurisdiction. The authors find important differences in nonpayment for defendants represented by a public defender relative to those represented by private counsel, including amounts assessed and the prevalence of full repayment. This paper uses novel data in an interesting way to address important gaps in the knowledge, and has a lot of potential. However, I have some concerns about its current version shared here that, to my read, prevent the paper from making its best contribution and the authors may wish to address as they continue to refine this scholarship.

            (1) My primary concern is that the paper is not well situated with the literature base. The literature review is very brief, only citing 7 works – none of which are peer-reviewed. Absent sufficient context, it’s difficult to assess the merits of this study, the analytic choices of the authors (e.g., sub analyses for court jurisdiction level and DWSL3 cases), and its contribution to the literature. Perhaps this brevity comes from the authors’ self-characterization of their paper as a “report” or the need to use substantial space to describe the intricacies of their data and method. However, it seems to me that a more in-depth literature view that engages with prior research and demonstrates how it advances scholarship is appropriate and warranted for this outlet. Could the authors revise the paper’s front end to more substantially engage with existing literatures to better define their work’s contribution?

            (2) It is clear that the authors took a meticulous approach to creating their analytic sample and measures from received data. However, I found the ‘Materials and Methods’ section to be in need of attention in two aspects.

            First, the high reliance on footnotes makes this section cumbersome to read and difficult to follow. This current structure ultimately left me with an unclear impression of the exact kinds of cases included in the data and the exact construction of measures. Further, some footnoted information seems to be too crucial to relegate there – for example, footnote i’s detail on the types of cases included in the data, an important factor in prior research on LFOs, seems worthy of discussion alongside the sample construction. Could some of this text be moved to the paper’s main body to make descriptions is easier to follow and give readers a better understanding of the data being analyzed? In addition, could the authors embed a table in this section that is devoted solely to presenting descriptive statistics of all variables included in the analysis, rather than presenting this data only for a subset of variables spread across tables? This would provide important context to the rest of the analysis (see point 3) and aid interpretation of the variable measures, should the authors choose to do so.

Second, the authors describe narrow inclusion criteria for case inclusion in their analytic sample in a thorough manner, doing a nice job of providing a reasonable and defensible rationale for their sample construction. There is not, however, a similarly detailed discussion of the limitations these decisions present for their analysis and conclusion. Such information seems crucial because the final sample is such a small subset – 15% – of the sample universe. This important (and appreciated) acknowledgement – “Authors acknowledge that trimming the dataset to this 133 extent may lead to loss of information, decreased statistical power, and weakened generalizability of findings.” – is worthy of expanded discussion. My concerns are less about the potential loss of statistical power. I’m more concerned about the implications of matters like using listwise deletion for cases with missing data on several important variables, excluding cases with multiple LFO assessment dates (presumably monthly fees for people on probation/supervision?), and excluding Seattle. Could the authors include a more systematic discussion on these fronts – if not in the ‘Materials and Methods’ section, perhaps in a separate ‘Limitations’ section towards the end of the manuscript?

            (3) What is the rationale for some of the statistical tests performed in the authors’ analysis? Could the authors provide reasons for their choice of methods and any important contextualizing information? For example, why are Wilcoxon rank sum tests used – because of the distribution of the financial variables? If so, please provide more information and/or justification.

            (4) The discussion section currently reads as a summary of the results – instead, could the authors consider using this space to connect their findings to prior research? A more thorough discussion of how their findings support or complicate existing research, what new is learned, and  the policy implications from these results in the particular courts in which they are observed would magnify the contribution of this paper.

Smaller Concerns:

In the Introduction, the authors reference findings from prior research on LFO debt and repayment but do not appropriately cite published work on these fronts.

Given the depth of inclusion criteria for this analysis, the authors may wish to use language different than “comprehensive” (pg. 1) to describe their analysis.

There seem to be grammar errors on lines 176 & 183.

There are a few important contextualizing details that seem to be missing, my apologies if I missed them: do data include restitution as an “other” LFO? Why is the median paid amount for indigent defendants missing from Table 1? How is ‘payment rate’ calculated and measured?

If space is an issue for the authors, they could cut much of the in-text description of the plots of regression results – much of this could be included in the Figure with a few additional labels.

Could the authors include substantive interpretation of regression coefficients to help the readers understand the magnitude of associations in context? In addition, could they consider using different terminology than “weak” correlations when describing regression coefficients? I think the use of “weak” speaks to the strength of the association as revealed by p-values, but the authors seem to be discussing the magnitude of the association.

Author Response

See PDF document attached.

Round 2

Reviewer 3 Report

I would like to commend the authors on their effort in this revised manuscript. The paper’s framing and context is much improved, and the authors have clearly taken care to include a more thorough discussion of their data, analytic design choices and limitations. I have some small remaining concerns that I share in this memo that the authors may wish to consider.

First, a big picture question. Can the authors comment on the reason for missing attorney data in 85,653 cases? Do you have a way to tease out if those cases are missing for ‘valid’ reasons (i.e., record keeping errors for cases that had an attorney present but it was not recorded for whatever reason), or are they missing because these persons did not have any legal representation while in court? The latter seems like a real possibility given the nature of cases processed in these courts, and likely includes people who fully repay on the day of assessment on their way out of court (speaking from personal experience with traffic tickets in municipal court). This non-representation is a different but nonetheless meaningful form of social advantage than private attorney. If possible to parse out, I wonder if these cases would follow a similar pattern to findings regarding private attorneys.

The most serious charge sub-analyses remain poorly motivated. Other than being the most prevalent charges in the dataset, why are they important to analyze with respect LFO assessment and repayment? What is it about these cases that merits their separate analysis to see if patterns conform with or diverge from sample-level patterns?

Some of the writing in Section 4.2.4 appears to be internally inconsistent. The authors say, “According to the models, assessment amounts have a modest impact on the rate of defendant’s LFO payment, with higher assessments being associated with lower rates of payment” then say “These findings dispel the notion that the reason defendants might have trouble paying their LFOs is simply due to their being assessed higher amounts.” The findings in this section seem to confirm that latter point exactly though, especially because they are derived from multivariable regression results. Assuming the authors distinguish their percentiles from low assessment amounts to higher assessment amounts, some revision that clarifies the authors’ assessment of their results is warranted.

The description of the missing data checks (lines 471-475) is quite vague. Can the authors be more specific in naming the tests used and noting how the data with incomplete data differ from the analytic sample? As written, I’m having a hard time seeing how the “data with incomplete information” differs from the “finale sample data” because both appear to suggest listwise deletion. More clarity on this important robustness check to support the rigor of their analysis and conclusions is warranted.

Smaller issues:

Grammar error on line 34-35 “issue”

The first clause of the sentence in lines 111-112 is a bit misleading in the context of the preceding review, and perhaps should be removed to focus on contributions from studying courts of limited jurisdiction in WA.

As currently written and placed, it is not clear which cases are referenced by the explanatory text of negative values in Figure 1.

Discussion of results on pages 10-11 could be enhanced with references back to the Figures conveying the regression results. This will help guide readers through the results presentation that is collapsed across different model estimations for each covariate.

Finally, the authors may want to include full regression output tables as supplemental text to the manuscript and the included Figures for ease of interpretation and specific numeric documentation of estimated associations.
